# Nonbacktracking Bounds on the Influence in Independent Cascade Models

**Emmanuel Abbe[1 2]  Sanjeev Kulkarni[2]  Eun Jee Lee[1]**

[1]Program in Applied and Computational Mathematics [2]The Department of Electrical Engineering
Princeton University
{eabbe, kulkarni, ejlee}@princeton.edu

## Abstract

This paper develops upper and lower bounds on the influence measure in a network, more precisely, the expected number of nodes that a seed set can influence in the independent cascade model. In particular, our bounds exploit nonbacktracking walks, Fortuin–Kasteleyn–Ginibre type inequalities, and are computed by message passing algorithms. Nonbacktracking walks have recently allowed for headways in community detection, and this paper shows that their use can also impact the influence computation. Further, we provide parameterized versions of the bounds that control the trade-off between the efficiency and the accuracy. Finally, the tightness of the bounds is illustrated with simulations on various network models.

## 1   Introduction

Influence propagation is concerned with the diffusion of information from initially influenced nodes, called *seeds*, in a network. Understanding how information propagates in networks has become a central problem in a broad range of fields, such as viral marketing [18], sociology [9, 20, 24], communication [13], epidemiology [21], and social network analysis [25].

One of the most fundamental questions on influence propagation is to estimate the *influence*, i.e. the expected number of influenced nodes at the end of the propagation given a set of seeds. Estimating the influence is central to diverse research problems related to influence propagation, such as the widely-known influence maximization problem — finding a set of $k$ nodes that maximizes the influence.

Recent studies on influence propagation have proposed various algorithms [12, 19, 4, 8, 23, 22] for the influence maximization problem while using Monte Carlo (MC) simulations to approximate the influence. The submodularity argument and the probabilistic error bound on MC give a probabilistic lower bound on the influence that is obtainable by the algorithms in terms of the true maximum influence. Despite its benefits on the influence maximization problem, approximating the influence via MC simulations is far from ideal for large networks; in particular, MC may require a large amount of computations in order to stabilize the approximation.

To overcome the limitations of Monte Carlo simulations, many researchers have been taking both algorithmic and theoretical approaches to approximate the influence of given seeds in a network. Chen and Teng [3] provided a probabilistic guarantee on estimating the influence of a single seed with a relative error bound with the expected running time $O(\ell(|V| + |E|)|V| \log |V|/\varepsilon^2)$, such that with probability $1 - 1/n^\ell$, for all node $v$, the computed influence of $v$ has relative error at most $\varepsilon$. Draief et al., [6] introduced an upper bound for the influence by using the spectral radius of the adjacency matrix. Tighter upper bounds were later suggested in [17] which relate the ratio of influenced nodes in a network to the spectral radius of the so-called *Hazard* matrix. Further, improved upper bounds which account for *sensitive* edges were introduced in [16].

In contrast, there has been little work on finding a tight lower bound for the influence. An exception is a work by Khim et al. [14], where the lower bound is obtained by only considering the influence through the *maximal-weighted* paths.

In this paper, we propose both upper and lower bounds on the influence using nonbacktracking walks and Fortuin–Kasteleyn–Ginibre (FKG) type inequalities. The bounds can be efficiently obtained by message passing implementation. This shows that nonbacktracking walks can also impact influence propagation, making another case for the use of nonbacktracking walks in graphical model problems as in [15, 10, 2, 1], discussed later in the paper. Further, we provide a parametrized version of the bounds that can adjust the trade-off between the efficiency and the accuracy of the bounds.

## 2 Background

We introduce here the independent cascade model and provide background for the main results.

**Definition 1** (Independent Cascade Model). *Consider a directed graph $G = (V, E)$ where $|V| = n$, a transmission probability matrix $\mathcal{P} \in [0, 1]^{n \times n}$, and a seed set $S_0 \subseteq V$. For all $u \in V$, let $N^+(u)$ be the set of out-neighbors of node $u$. The independent cascade model $IC(G, \mathcal{P}, S_0)$ sequentially generates the influenced set $S_t \subseteq V$ for each discrete time $t \geq 1$ as follows. At time $t$, $S_t$ is initialized to be an empty set. Then, each node $u \in S_{t-1}$ attempts to influence $v \in N^+(u) \setminus \cup_{i=0}^{t-1} S_i$ with probability $\mathcal{P}_{uv}$, i.e. node $u$ influences its uninfluenced out-neighbor $v$ with probability $\mathcal{P}_{uv}$. If $v$ is influenced at time $t$, add $v$ to $S_t$. The process stops at $T$ if $S_T = \emptyset$ at the end of the step $t = T$. The set of the influenced nodes at the end of propagation is defined as $S = \cup_{i=0}^{T-1} S_t$.*

We often refer an edge $(u, v)$ being *open* if node $u$ influences node $v$. The IC model is equivalent to the live-arc graph model, where the influence happens at once, rather than sequentially. The live-arc graph model first decides the state of every edge with a Bernoulli trial, i.e. edge $(u, v)$ is open independently with probability $\mathcal{P}_{uv}$ and closed, otherwise. Then, the set of influenced nodes is defined as the nodes that are reachable from at least one of the seeds by the open edges.

**Definition 2** (Influence). *The expected number of nodes that are influenced at the end of the propagation process is called the* influence *(rather than the expected influence, with a slight abuse of terminology) of $IC(G, \mathcal{P}, S_0)$, and is defined as*

$$\sigma(S_0) \quad = \quad \sum_{v \in V} \mathbb{P}(v \text{ is influenced}). \tag{1}$$

It is shown in [5] that computing the influence $\sigma(S_0)$ in the independent cascade model $IC(G, \mathcal{P}, S_0)$ is #P-hard, even with a single seed, i.e. $|S_0| = 1$.

Next, we define nonbacktracking (NB) walks on a directed graph. Nonbacktracking walks have already been used for studying the characteristics of networks. To the best of our knowledge, the use of NB walks in the context of epidemics was first introduced in the paper of Karrer et al. [11] and later applied to percolation in [10]. In particular, Karrer et al. reformulate the spread of influence as a message passing process and demonstrate how the resulting equations can be used to calculate an upper bound on the number of nodes that are susceptible at a given time. As we shall see, we take a different approach to the use of the NB walks, which focuses on the effective contribution of a node in influencing another node and accumulates such contributions to obtain upper and lower bounds. More recently, nonbacktracking walks are used for community detection [15, 2, 1].

**Definition 3** (Nonbacktracking Walk). *Let $G = (V, E)$ be a directed graph. A nonbacktracking walk of length $k$ is defined as $w^{(k)} = (v_0, v_1, \ldots, v_k)$, where $v_i \in V$ and $(v_{i-1}, v_i) \in E$ for all $i \in [k]$, and $v_{i-1} \neq v_{i+1}$ for all $i \in [k-1]$.*

We next recall a key inequality introduced by Fortuin et. al [7].

**Theorem 1** (FKG Inequality). *Let $(\Gamma, \prec)$ be a distributive lattice, where $\Gamma$ is a finite partially ordered set, ordered by $\prec$, and let $\mu$ be a positive measure on $\Gamma$ satisfying the following condition: for all $x, y \in \Gamma$,*

$$\mu(x \wedge y)\mu(x \vee y) \quad \geq \quad \mu(x)\mu(y),$$

*where $x \wedge y = \max\{z \in \Gamma : z \preceq x, z \preceq y\}$ and $x \vee y = \min\{z \in \Gamma : y \preceq z, y \preceq z\}$. Let $f$ and $g$ be both increasing (or both decreasing) functions on $\Gamma$. Then,*

$$\left(\sum_{x \in \Gamma} \mu(x)\right)\left(\sum_{x \in \Gamma} f(x)g(x)\mu(x)\right) \quad \geq \quad \left(\sum_{x \in \Gamma} f(x)\mu(x)\right)\left(\sum_{x \in \Gamma} g(x)\mu(x)\right). \tag{2}$$

FKG inequality is instrumental in studying influence propagation since the probability that a node is influenced is nondecreasing with respect to the partial order of random variables describing the states, open or closed, of the edges.

## 3 Nonbacktracking bounds on the influence

In this section, we present upper and lower bounds on the influence in the independent cascade model and explain the motivations and intuitions of the bounds. The bounds utilize nonbacktracking walks and FKG inequalities and are computed efficiently by message passing algorithms. In particular, the upper bound on a network based on a graph $G(V, E)$ runs in $O(|V|^2 + |V||E|)$ and the lower bound runs in $O(|V| + |E|)$, whereas Monte Carlo simulation would require $O(|V|^3 + |V|^2|E|)$ computations without knowing the variance of the influence, which is harder to estimate than the influence. The reason for the large computational complexity of MC is that in order to ensure that the standard error of the estimation does not grow with respect to $|V|$, MC requires $O(|V|^2)$ computations. Hence, for large networks, where MC may not be feasible, our algorithms can still provide bounds on the influence.

Furthermore, from the proposed upper $\sigma^+$ and lower bounds $\sigma^-$, we can compute an upper bound on the variance given by $(\sigma^+ - \sigma^-)^2/4$. This could be used to estimate the number of computations needed by MC. Computing the upper bound on the variance with the proposed bounds can be done in $O(|V|^2 + |V||E|)$, whereas computing the variance with MC simulation requires $O(|V|^5 + |V|^4|E|)$.

### 3.1 Nonbacktracking upper bounds (NB-UB)

We start by defining the following terms for the independent cascade model $IC(G, \mathcal{P}, S_0)$, where $G = (V, E)$ and $|V| = n$.

**Definition 4.** *For any $v \in V$, we define the set of in-neighbors $N^-(v) = \{u \in V : (u, v) \in E\}$ and the set of out-neighbors $N^+(v) = \{u \in V : (v, u) \in E\}$.*

**Definition 5.** *For any $v \in V$ and $l \in [n-1]$, the set $P_l(S_0 \to v)$ is defined as the set of all paths with length $l$ from any seed $s \in S_0$ to $v$. We call a path $P$ is* open *iff every edge in $P$ is open. For $l = 0$, we define $P_0(S_0 \to v)$ as the set (of size one) of the zero-length path containing node $v$ and assume the path $P \in P_0(S_0 \to v)$ is open iff $v \in S_0$.*

**Definition 6.** *For any $v \in V$ and $l \in \{0, \ldots, n-1\}$, we define*

$$
\begin{align}
p(v) &= \mathbb{P}(v \text{ is influenced}) \tag{3}\\
p_l(v) &= \mathbb{P}(\cup_{P \in P_l(S_0 \to v)}\{P \text{ is open}\}) \tag{4}\\
p_l(u \to v) &= \mathbb{P}(\cup_{P \in P_l(S_0 \to u), P \not\ni v}\{P \text{ is open and edge } (u, v) \text{ is open}\}) \tag{5}
\end{align}
$$

In other words, $p_l(v)$ is the probability that node $v$ is influenced by open paths of length $l$, i.e. there exists an open path of length $l$ from a seed to $v$, and $p_l(u \to v)$ is the probability that $v$ is influenced by node $u$ with open paths of length $l+1$, i.e. there exists an open path of length $l+1$ from a seed to $v$ that ends with edge $(u, v)$.

**Lemma 1.** *For any $v \in V$,*

$$
p(v) \leq 1 - \prod_{l=0}^{n-1}(1 - p_l(v)). \tag{6}
$$

*For any $v \in V$ and $l \in [n-1]$,*

$$
p_l(v) \leq 1 - \prod_{u \in N^-(v)}(1 - p_{l-1}(u \to v)). \tag{7}
$$

Lemma 1, which can be proved by FKG inequalities, suggests that given $p_{l-1}(u \to v)$, we may compute an upper bound on the influence. Ideally, $p_{l-1}(u \to v)$ can be computed by considering all paths that end with $(u, v)$ having length $l$. However, this results in exponential complexity $O(n^l)$, as $l$ goes up to $n-1$. Thus, we present an efficient way to compute an upper bound $\text{UB}_{l-1}(u \to v)$ on $p_{l-1}(u \to v)$, which in turns gives an upper bound $\text{UB}_l(v)$ on $p_l(v)$, with the following recursion formula.

**Definition 7.** *For all $l \in \{0, \dots, n-1\}$ and $u, v \in V$ such that $(u,v) \in E$, $\mathrm{UB}_l(u) \in [0,1]$ and $\mathrm{UB}_l(u \to v) \in [0,1]$ are defined recursively as follows.*
*Initial condition: For every $s \in S_0$, $s^+ \in N^+(s)$, $u \in V \setminus S_0$, and $v \in N^+(u)$,*

$$\mathrm{UB}_0(s) = 1, \ \mathrm{UB}_0(s \to s^+) = \mathcal{P}_{ss^+} \tag{8}$$

$$\mathrm{UB}_0(u) = 0, \ \mathrm{UB}_0(u \to v) = 0. \tag{9}$$

*Recursion: For every $l \in [n-1]$, $s \in S_0$, $s^+ \in N^+(s)$, $s^- \in N^-(s)$, $u \in V \setminus S_0$, and $v \in N^+(u) \setminus S_0$,*

$$\mathrm{UB}_l(s) = 0, \mathrm{UB}_l(s \to s^+) = 0, \mathrm{UB}_l(s^- \to s) = 0 \tag{10}$$

$$\mathrm{UB}_l(u) = 1 - \prod_{w \in N^-(u)} (1 - \mathrm{UB}_{l-1}(w \to u)) \tag{11}$$

$$\mathrm{UB}_l(u \to v) = \begin{cases} \mathcal{P}_{uv}(1 - \frac{1 - \mathrm{UB}_l(u)}{1 - \mathrm{UB}_{l-1}(v \to u)}), & if \ v \in N^-(u) \\ \mathcal{P}_{uv}\mathrm{UB}_l(u), & otherwise. \end{cases} \tag{12}$$

Equation (10) follows from that for any seed node $s \in S_0$ and for all $l > 0$, the probabilities $p_l(s) = 0$, $p_l(s \to s^+) = 0$, and $p_l(s^- \to s) = 0$. A naive way to compute $\mathrm{UB}_l(u \to v)$ is $\mathrm{UB}_l(u \to v) = \mathcal{P}_{uv}\mathrm{UB}_{l-1}(u)$, but this results in an extremely loose bound due to the backtracking. For a tighter bound, we use nonbacktracking in Equation (12), i.e. when computing $\mathrm{UB}_l(u \to v)$, we ignore the contribution of $\mathrm{UB}_{l-1}(v \to u)$.

**Theorem 2.** *For any independent cascade model $IC(G, \mathcal{P}, S_0)$,*

$$\sigma(S_0) \ \leq \ \sum_{v \in V}(1 - \prod_{l=0}^{n-1}(1 - \mathrm{UB}_l(v))) =: \sigma^+(S_0), \tag{13}$$

*where $\mathrm{UB}_l(v)$ is obtained recursively as in Definition 7.*

Next, we present Nonbacktracking Upper Bound (NB-UB) algorithm which computes $\mathrm{UB}_l(v)$ and $\mathrm{UB}_l(u \to v)$ by message passing. At the $l$-th iteration, the variables in NB-UB represent as follows.

- $\cdot$ $S_l$ is the set of nodes that are processed at the $l$-th iteration.
- $\cdot$ $\mathrm{M}_{\mathrm{curr}}(v) = \{(u, \mathrm{UB}_{l-1}(u \to v)) : u \text{ is an in-neighbor of } v, \text{ and } u \in S_{l-1}\}$ is the set of pairs (previously processed in-neighbor $u$ of $v$, incoming message from $u$ to $v$).
- $\cdot$ $\mathrm{MSrc}(v) = \{u : u \text{ is a in-neighbor of } v, \text{ and } u \in S_{l-1}\}$ is the set of in-neighbor nodes of $v$ that were processed at the previous step.
- $\cdot$ $\mathrm{M}_{\mathrm{curr}}(v)[u] = \mathrm{UB}_{l-1}(u \to v)$ is the incoming message from $u$ to $v$.
- $\cdot$ $\mathrm{M}_{\mathrm{next}}(v) = \{(u, \mathrm{UB}_l(u \to v)) : u \text{ is an in-neighbor of } v, \text{ and } u \in S_l\}$ is the set of pairs (currently processed in-neighbor $u$, next iteration's incoming message from $u$ to $v$).

---

**Algorithm 1** Nonbacktracking Upper Bound (NB-UB)

---

**Initialize:** $\mathrm{UB}_l(v) = 0$ for all $0 \leq l \leq n-1$ and $v \in V$
**Initialize:** Insert $(s, 1)$ to $\mathrm{M}_{\mathrm{next}}(s)$ for all $s \in S_0$
**for** $l = 0$ **to** $n-1$ **do**
    **for** $u \in S_l$ **do**
        $\mathrm{M}_{\mathrm{curr}}(u) = \mathrm{M}_{\mathrm{next}}(u)$ and Clear $\mathrm{M}_{\mathrm{next}}(u)$
        $\mathrm{UB}_l(u) = \texttt{ProcessIncomingMsg}_{\mathrm{UB}}(\mathrm{M}_{\mathrm{curr}}(u))$
    **for** $u \in S_l$ **do**
        **for** $v \in N^+(u) \setminus S_0$ **do**
            $S_{l+1}.\mathrm{insert}(v)$
            **if** $v \in \mathrm{MSrc}(u)$ **then**
                $\mathrm{UB}_l(u \to v) = \texttt{GenerateOutgoingMsg}_{\mathrm{UB}}(\mathrm{M}_{\mathrm{curr}}(u)[v], \mathrm{UB}_l(u), \mathcal{P}_{uv})$
                $\mathrm{M}_{\mathrm{next}}(v).\mathrm{insert}((u, \mathrm{UB}_l(u \to v)))$.
            **else**
                $\mathrm{UB}_l(u \to v) = \texttt{GenerateOutgoingMsg}_{\mathrm{UB}}(0, \mathrm{UB}_l(u), \mathcal{P}_{uv})$
                $\mathrm{M}_{\mathrm{next}}(v).\mathrm{insert}((u, \mathrm{UB}_l(u \to v)))$.
**Output:** $\mathrm{UB}_l(u)$ for all $l, u$

---

At the beginning, every seed node $s \in S_0$ is initialized such that $\mathrm{M_{curr}}(s) = \{(s, 1)\}$ in order to satisfy the initial condition, $\mathrm{UB}_0(s) = 1$. For each $l$-th iteration, every node $u$ in $S_l$ is processed as follows. First, `ProcessIncomingMsg`$_{\mathrm{UB}}$($\mathrm{M_{curr}}(u)$) computes $\mathrm{UB}_l(u)$ as in Equation (11). Second, $u$ passes a message to its neighbor $v \in N^+(u) \setminus S_0$ along the edge $(u, v)$, and $v$ stores (inserts) the message in $\mathrm{M_{next}}(v)$ for the next iteration. The message contains 1) the source of the message, $u$, and 2) $\mathrm{UB}_l(u \to v)$, which is computed as in Equation (12), by the function `GenerateOutgoingMsg`$_{\mathrm{UB}}$. Finally, the algorithm outputs $\mathrm{UB}_l(u)$ for all $u \in V$ and $l \in \{0, \ldots, n-1\}$, and the upper bound $\sigma^+(S_0)$ is computed by Equation (13). The description of how the algorithm runs on a small network can be found in the supplementary material.

**Computational complexity**: Notice that for each iteration $l \in \{0, \ldots, n-1\}$, the algorithm accesses at most $n$ nodes, and for each node $v$, the functions `ProcessIncomingMsg`$_{\mathrm{UB}}$ and `GenerateOutgoingMsg`$_{\mathrm{UB}}$ are computed in $O(\deg(v))$ and $O(1)$, respectively. Therefore, the worst case computational complexity is $O(|V|^2 + |V||E|)$.

### 3.2 Nonbacktracking lower bounds (NB-LB)

A naive way to compute a lower bound on the influence in a network $IC(G, \mathcal{P}, S_0)$ is to reduce the network to a (spanning) tree network, by removing edges. Then, since there is a unique path from a node to another, we can compute the influence of the tree network, which is a lower bound on the influence in the original network, in $O(|V|)$. We take this approach of generating a subnetwork from the original network, yet we avoid the significant gap between the bound and the influence by considering the following directed acyclic subnetwork, in which there is no backtracking walk.

**Definition 8** (Min-distance Directed Acyclic Subnetwork). *Consider an independent cascade model $IC(G, \mathcal{P}, S_0)$ with $G = (V, E)$ and $|V| = n$. Let $d(S_0, v) := \min_{s \in S_0} d(s, v)$, i.e. the minimum distance from a seed in $S_0$ to $v$. A minimum-distance directed acyclic subnetwork (MDAS), $IC(G', \mathcal{P}', S_0)$, where $G' = (V', E')$, is obtained as follows.*

- *$V' = \{v_1, ..., v_n\}$ is an ordered set of nodes such that $d(S_0, v_i) \leq d(S_0, v_j)$, for every $i < j$.*
- *$E' = \{(v_i, v_j) \in E : i < j\}$, i.e. $E'$ is obtained from $E$ by removing edges whose source node comes later in the order than its destination node.*
- *$\mathcal{P}'_{v_i v_j} = \mathcal{P}_{v_i v_j}$, if $(v_i, v_j) \in E'$, and $\mathcal{P}'_{v_i v_j} = 0$, otherwise.*

If there are multiple ordered sets of vertices satisfying the condition, we may choose one arbitrarily.

For any $k \in [n]$, let $p(v_k)$ be the probability that $v_k \in V'$ is influenced in the MDAS, $IC(G', \mathcal{P}', S_0)$. Since $p(v_k)$ is equivalent to the probability of the union of the events that an in-neighbor $u_i \in N^-(v_k)$ influences $v_k$, $p(v_k)$ can be computed by the principle of inclusion and exclusion. Thus, we may compute a lower bound on $p(v_k)$, using Bonferroni inequalities, if we know the probabilities that in-neighbors $u$ and $v$ both influences $v_k$, for every pair $u, v \in N^-(v_k)$. However, computing such probabilities can take $O(k^k)$. Hence, we present $\mathrm{LB}(v_k)$ which efficiently computes a lower bound on $p(v_k)$ by the following recursion.

**Definition 9.** *For all $v_k \in V'$, $\mathrm{LB}(v_k) \in [0, 1]$ is defined by the recursion on $k$ as follows.*
Initial condition: *For every $v_s \in S_0$,*

$$\mathrm{LB}(v_s) = 1. \tag{14}$$

Recursion: *For every $v_k \in V' \setminus S_0$,*

$$\mathrm{LB}(v_k) = \sum_{i=1}^{m^*} \left( \mathcal{P}'_{u_i v_k} \mathrm{LB}(u_i) \left(1 - \sum_{j=1}^{i-1} \mathcal{P}'_{u_j v_k}\right) \right), \tag{15}$$

*where $N^-(v_k) = \{u_1, \ldots, u_m\}$ is the ordered set of in-neighbors of $v_k$ in $IC(G', \mathcal{P}', S_0)$ and $m^* = \max\{m' \leq m : \sum_{j=1}^{m'-1} \mathcal{P}'_{u_j v_k} \leq 1\}$.*

**Remark.** Since the $i$-th summand in Equation (15) can utilize $\sum_{j=1}^{i-2} \mathcal{P}'_{u_j v_k}$, which is already computed in $(i-1)$-th summand, to compute $\sum_{j=1}^{i-1} \mathcal{P}'_{u_j v_k}$, the summation takes at most $O(\deg(v_k))$.

**Theorem 3.** *For any independent cascade model $IC(G, \mathcal{P}, S_0)$ and its MDAS $IC(G', \mathcal{P}', S_0)$,*

$$\sigma(S_0) \geq \sum_{v_k \in V'} \mathrm{LB}(v_k) =: \sigma^-(S_0), \tag{16}$$

*where $\mathrm{LB}(v_k)$ is obtained recursively as in Definition 9.*

Next, we present Nonbacktracking Lower Bound (NB-LB) algorithm which efficiently computes $\text{LB}(v_k)$. At the $k$-th iteration, the key variable in NB-LB has the following meaning.

· $\text{M}(v_k) = \{(\text{LB}(v_j), \mathcal{P}'_{v_j v_k}) : v_j \text{ is an in-neighbor of } v_k\}$ is the set of pairs (incoming message from an in-neighbor $v_j$ to $v_k$, the transmission probability of edge $(v_j, v_k)$).

---

**Algorithm 2** Nonbacktracking Lower Bound (NB-LB)

---

**Input:** directed acyclic network $IC(G', \mathcal{P}', S_0)$
**Initialize:** $\sigma^- = 0$
**Initialize:** Insert $(1, 1)$ to $\text{M}(v_i)$ for all $v_i \in S_0$
**for** $k = 1$ **to** $n$ **do**
    $\text{LB}(v_k) = \texttt{ProcessIncomingMsg}_{\text{LB}}(\text{M}(v_k))$
    $\sigma^- \mathrel{+}= \text{LB}(v_k)$
    **for** $v_l \in N^+(v_k) \setminus S_0$ **do**
        $\text{M}(v_l).\text{insert}((\text{LB}(v_k), \mathcal{P}'_{v_k v_l}))$
**Output:** $\sigma^-$

---

At the beginning, every seed node $s \in S_0$ is initialized such that $\text{M}(s) = \{(1, 1)\}$ in order to satisfy the initial condition, $\text{LB}(s) = 1$. For each $k$-th iteration, node $v_k$ is processed as follows. First, $\text{LB}(v_k)$ is computed as in the Equation (15), by the function $\texttt{ProcessIncomingMsg}_{\text{LB}}$, and added to $\sigma^-$. Second, $v_k$ passes the message $(\text{LB}(v_k), \mathcal{P}'_{v_k v_l})$ to its out-neighbor $v_l \in N^+(v_k) \setminus S_0$, and $v_l$ stores (inserts) it in $\text{M}(v_l)$. Finally, the algorithm outputs $\sigma^-$, the lower bound on the influence. The description of how the algorithm runs on a small network can be found in the supplementary material.

**Computational complexity:** Obtaining an arbitrary directed acyclic subnetwork from the original network takes $O(|V| + |E|)$. Next, the algorithm iterates through the nodes $V' = \{v_1, \ldots, v_n\}$. For each node $v_k$, $\texttt{ProcessIncomingMsg}_{\text{LB}}$ takes $O(\deg(v_k))$, and $v_k$ sends messages to its out-neighbors in $O(\deg(v_k))$. Hence, the worst case computational complexity is $O(|V| + |E|)$.

### 3.3 Tunable bounds

In this section, we briefly introduce the parametrized version of NB-UB and NB-LB which provide control to adjust the trade-off between the efficiency and the accuracy of the bounds.

**Upper bounds (tNB-UB):** Given a non-negative integer $t \leq n - 1$, for every node $u \in V$, we compute the probability $p_{\leq t}(u)$ that node $u$ is influenced by open paths whose length is less than or equal to $t$, and for each $v \in N^+(u)$, we compute the probability $p_t(u \rightarrow v)$. Then, we start NB-UB from $l = t + 1$ with the new initial conditions that $\text{UB}_t(u \rightarrow v) = p_t(u \rightarrow v)$ and $\text{UB}_t(u) = p_{\leq t}(u)$, and compute the upper bound as $\sum_{v \in V}(1 - \prod_{l=t}^{n-1}(1 - \text{UB}_l(v)))$.

For higher values of $t$, the algorithm results in tighter upper bounds, while the computational complexity may increase exponentially for dense networks. Thus, this method is most applicable in sparse networks, where the degree of each node is bounded.

**Lower bounds (tNB-LB):** We first order the set of nodes $\{v_1, \ldots, v_n\}$ such that $d(S_0, v_i) \leq d(S_0, v_j)$ for every $i < j$. Given a non-negative integer $t \leq n$, we obtain a subnetwork $IC(G[V_t], \mathcal{P}[V_t], S_0 \cap V_t)$ of size $t$, where $G[V_t]$ is the subgraph induced by the set of nodes $V_t = \{v_1, \ldots, v_t\}$, and $\mathcal{P}[V_t]$ is the corresponding transmission probability matrix. For each $v_i \in V_t$, we compute the exact probability $p_t(v_i)$ that node $v_i$ is influenced in the subnetwork $IC(G[V_t], \mathcal{P}[V_t], S_0 \cap V_t)$. Then, we start NB-LB from $i = t + 1$ with the new initial conditions that $\text{LB}(v_k) = p_t(v_k)$, for all $k \leq t$.

For larger $t$, the algorithm results in tighter lower bounds. However, the computational complexity may increase exponentially with respect to $t$, the size of the subnetwork. This algorithm can adopt Monte Carlo simulations on the subnetwork to avoid the large computational complexity. However, this modification results in probabilistic lower bounds, rather than theoretically guaranteed lower bounds. Nonetheless, this can still give a significant improvement, because the Monte Carlo simulations on a smaller size of network require less computation to stabilize the estimation.

# 4 Experimental Results

In this section, we evaluate the NB-UB and NB-LB in independent cascade models on a variety of classical synthetic networks.

**Network Generation**. We consider $4$ classical random graph models with the parameters shown as follows: Erdos Renyi random graphs with $ER(n = 1000, p = 0.003)$, scale-free networks $SF(n = 1000, \alpha = 2.5)$, random regular graphs $Reg(n = 1000, d = 3)$, and random tree graphs with power-law degree distributions $T(n = 1000, \alpha = 3)$. For each graph model, we generate $100$ networks $IC(G, pA, \{s\})$ as follows. The graph $G$ is the largest connected component of a graph drawn from the graph model, the seed node $s$ is a randomly selected vertex, and $A$ is the adjacency matrix of $G$. The corresponding IC model has the same transmission probability $p$ for every edge.

**Evaluation of Bounds**. For each network generated, we compute the following quantities for each $p \in \{0.1, 0.2, \dots, 0.9\}$.

· $\sigma_{mc}$: the estimation of the influence with $10^6$ Monte Carlo simulations.

· $\sigma^+$: the upper bound obtained by NB-UB.

· $\sigma_{spec}^+$: the spectral upper bound by [17].

· $\sigma^-$: the lower bound obtained by NB-LB.

· $\sigma_{prob}^-$: the probabilistic lower bound obtained by 10 Monte Carlo simulations.

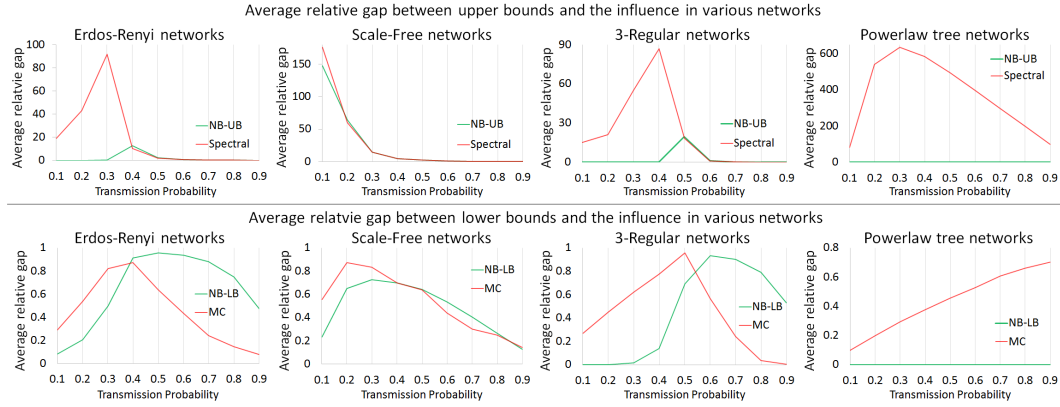

Figure 1: This figure compares the average relative gap of the bounds: NB-UB, the spectral upper bound in [17], NB-LB, and the probabilistic lower bound computed by MC simulations, for various types of networks.

The probabilistic lower bound is chosen for the experiments since there has not been any tight lower bound. The sample size of $10$ is determined to overly match the computational complexity of NB-LB algorithm. In Figure 1, we compare the average relative gap of the bounds for every network model and for each transmission probability, where the true value is assumed to be $\sigma_{mc}$. For example, the average relative gap of NB-UB for 100 Erdos Renyi networks $\{\mathcal{N}_i\}_{i=1}^{100}$ with the transmission probability $p$ is computed by $\frac{1}{100} \sum_{i \in [100]} \frac{\sigma^+[\mathcal{N}_i] - \sigma_{mc}[\mathcal{N}_i]}{\sigma_{mc}[\mathcal{N}_i]}$, where $\sigma^+[\mathcal{N}_i]$ and $\sigma_{mc}[\mathcal{N}_i]$ denote the NB-UB and the MC estimation, respectively, for the network $\mathcal{N}_i$.

**Results**. Figure 1 shows that NB-UB outperforms the upper bound in [17] for the Erdos-Renyi and random 3-regular networks, and performs comparably for the scale-free networks. Also, NB-LB gives tighter bounds than the MC bounds on the Erdos-Renyi, scale-free, and random regular networks when the transmission probability is small, $p < 0.4$. Both NB-UB and NB-LB compute the exact influence for the tree networks since both algorithms avoid backtracking walks.

Next, we show the bounds on exemplary networks.

## 4.1 Upper Bounds

**Selection of Networks**. In order to illustrate a typical behavior of the bounds, we have chosen the network in Figure 2a as follows. First, we generate 100 random 3-regular graphs $G$ with 1000 nodes and assign a random seed $s$. Then, the corresponding IC model is defined as $IC(G, \mathcal{P} =$

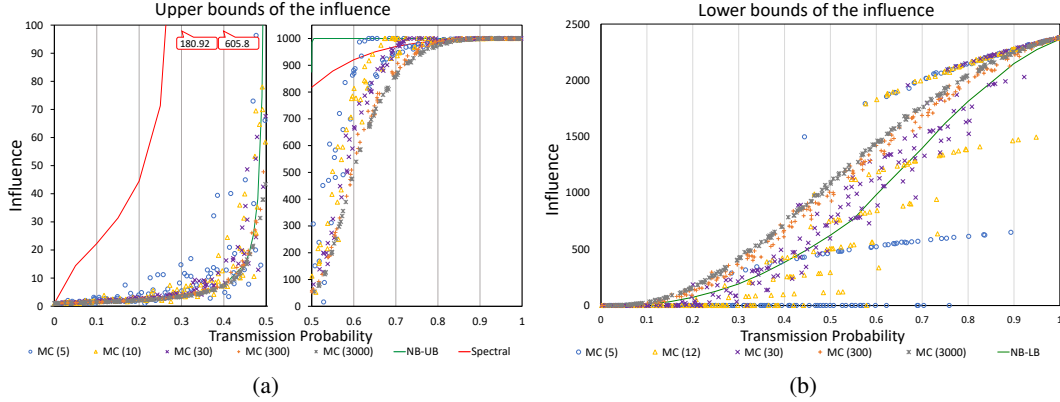

Figure 2: (a) The figure compares various upper bounds on the influence in the 3-regular network in section 4.1. The MC upper bounds are computed with various simulation sizes and shown with the data points indicated with $\mathrm{MC}(N)$, where $N$ is the number of simulations. The spectral upper bound in [17] is shown in red line, and NB-UB is shown in green line.

(b) The figure shows lower bounds on the influence of a scale-free network in section 4.2. The probabilistic lower bounds shown with points are obtained from Monte Carlo simulations with various simulation sizes, and the data points indicated with $\mathrm{MC}(N)$ are obtained by $N$ number of simulations. NB-LB is shown in green line.

$pA, S_0 = \{s\}$). For each network, we compute NB-UB and MC estimation. Then, we compute the score for each network, where the score is defined as the sum of the square differences between the upper bounds and MC estimations over the transmission probability $p \in \{0.1, 0.2, \ldots, 0.9\}$. Finally, a graph whose score is the median from all 100 scores is chosen for Figure 2a.

**Results**. In figure 2a, we compare 1) the upper bounds introduced [17] and 2) the probabilistic upper bounds obtained by Monte Carlo simulations with 99% confidence level, to NB-UB. The MC upper bounds are computed with the various sample sizes $N \in \{5, 10, 30, 300, 3000\}$. It is evident from the figure that a larger sample size provides a tighter probabilistic upper bound. NB-UB outperforms the bound by [17] and the probabilistic MC bound when the transmission probability is relatively small. Further, it shows a similar trend as the MC simulations with a large sample size.

## 4.2 Lower Bounds

**Selection of Networks**. We adopt a similar selection process as in the selection for the upper bounds, but with the scale free networks, with 3000 nodes and $\alpha = 2.5$.

**Results**. We compare probabilistic lower bounds obtained by MC with 99% confidence level to NB-LB. The lower bounds from Monte Carlo simulations are computed with various sample sizes $N \in \{5, 12, 30, 300, 3000\}$, which accounts for a constant, $\log(|V|)$, $0.01|V|$, $0.1|V|$, and $|V|$. NB-LB outperforms the probabilistic bounds by MC with small sample sizes. Recall that the computational complexity of the lower bound in algorithm 2 is $O(|V| + |E|)$, which is the computational complexity of a constant number of Monte Carlo simulations. In figure 2b, it shows that NB-LB is tighter than the probabilistic lower bounds with the same computational complexity, and it also agrees with the behavior of the MC simulations.

## 5 Conclusion

In this paper, we propose both upper and lower bounds on the influence in the independent cascade models and provide algorithms to efficiently compute the bounds. We extend the results by proposing tunable bounds which can adjust the trade-off between the efficiency and the accuracy. Finally, the tightness and the performance of the bounds are shown with the experimental results. One can further improve the bounds considering $r$-nonbacktracking walks, i.e. avoiding cycles of length $r$ rather than just backtracks, and we leave this for future study.

**Acknowledgement**. The authors thank Colin Sandon for helpful discussions. This research was partly supported by the NSF CAREER Award CCF-1552131 and the ARO grant W911NF-16-1-0051

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
