[Supplementary Material · NIPS17_sup_0131.pdf]

# Supplementary material

## 1 Illustrations of the algorithms

### 1.1 Nonbacktracking upper bounds (NB-UB)

Figure 1: The step-wise illustration of NB-UB algorithm on the example network.

| | $l=0$ | | $l=1$ | | $l=2$ | | $l=3$ | |
|---|---|---|---|---|---|---|---|---|
| $S_l$ | $\{b\}$ | | $\{a,c\}$ | | $\{d\}$ | | $\{a,c,e\}$ | |
| | $M_{curr}$ | $UB_0$ | $M_{curr}$ | $UB_1$ | $M_{curr}$ | $UB_2$ | $M_{curr}$ | $UB_3$ |
| $a$ | $\varnothing$ | $0$ | $\{(b,p)\}$ | $p$ | $\varnothing$ | $0$ | $\{(d,p^3)\}$ | $p^3$ |
| $b$ | $\{(b,1)\}$ | $1$ | $\varnothing$ | $0$ | $\varnothing$ | $0$ | $\varnothing$ | $0$ |
| $c$ | $\varnothing$ | $0$ | $\{(b,p)\}$ | $p$ | $\varnothing$ | $0$ | $\{(d,p^3)\}$ | $p^3$ |
| $d$ | $\varnothing$ | $0$ | $\varnothing$ | $0$ | $\{(a,p^2),(c,p^2)\}$ | $2p^2-p^4$ | $\varnothing$ | $0$ |
| $e$ | $\varnothing$ | $0$ | $\varnothing$ | $0$ | $\varnothing$ | $0$ | $\{(d,2p^3-p^5)\}$ | $2p^3-p^5$ |
| Out -Prob | $UB_0(b{\to}a)=p$ $UB_0(b{\to}c)=p$ | | $UB_1(a{\to}d)=p^2$ $UB_1(c{\to}d)=p^2$ | | $UB_2(d{\to}a)=p^3$ $UB_2(d{\to}c)=p^3$ $UB_2(d{\to}e)=2p^3-p^5$ | | $UB_3(a{\to}d)=0$ $UB_3(c{\to}d)=0$ $UB_3(e{\to}d)=0$ | |

Table 1: The values of the key variables in NB-UB algorithm on the example network in Figure 1.

In Figure 1, we describe how NB-UB algorithm runs on a small independent cascade model $IC(G,\mathcal{P},S_0)$ defined on an undirected graph $G=(V,E)$, where $V=\{a,b,c,d\}$, $S_0=\{b\}$, and every edge has the same transmission probability $p$. For each $l$, Table 1 shows the values of the key variables, $S_l$, $M_{curr}$, and $UB_l$, in the algorithm and $UB_l(u{\to}v)$ for every pair $u,v$ such that $u\in S_l$ and $v\in N^+(u)\setminus S_0$.

For example, at $l=2$, since $S_2=\{d\}$, node $d$ is processed. Recall that, at $l=1$, node $a$ sent the message $(a,UB_1(a{\to}d))$ to $d$, and node $c$ sent the message $(c,UB_1(c{\to}d))$ to $d$. Thus,

$$M_{curr}(d) = \{(a,UB_1(a{\to}d)),(c,UB_1(c{\to}d))\}=\{(a,p^2),(c,p^2)\} \tag{1}$$

$$MSrc(d) = \{a,c\}, \tag{2}$$

and node $d$ is processed as follows. First, compute $UB_2(d)$ as

$$UB_2(d) = \texttt{ProcessIncomingMsg}_{UB}(M_{curr}(d)) \tag{3}$$

$$= 1-(1-UB_1(a{\to}d))(1-UB_1(c{\to}d))=2p^2-p^4. \tag{4}$$

Next, set $S_3=N^+(d)\setminus S_0=\{a,c,e\}$ and compute the following messages. Since $a,c\in MSrc(d)$ and $e\notin MSrc(d)$,

$$UB_2(d{\to}a) = \texttt{GenerateOutgoingMsg}_{UB}(UB_1(a{\to}d),UB_2(d),\mathcal{P}_{da}) \tag{5}$$

$$= \mathcal{P}_{da}(1-\frac{1-UB_2(d)}{1-UB_1(a{\to}d)})=p^3 \tag{6}$$

$$UB_2(d{\to}c) = \texttt{GenerateOutgoingMsg}_{UB}(UB_1(c{\to}d),UB_2(d),\mathcal{P}_{dc}) \tag{7}$$

$$= \mathcal{P}_{dc}(1-\frac{1-UB_2(d)}{1-UB_1(c{\to}d)})=p^3 \tag{8}$$

$$UB_2(d{\to}e) = \texttt{GenerateOutgoingMsg}_{UB}(0,UB_2(d),\mathcal{P}_{de}) \tag{9}$$

$$= \mathcal{P}_{de}(1-\frac{1-UB_2(d)}{1-0})=\mathcal{P}_{de}UB_2(d)=2p^3-p^5. \tag{10}$$

Then, node $d$ send messages $(d,UB_2(d{\to}a))$ to $a$, $(d,UB_2(d{\to}c))$ to $c$, and $(d,UB_2(d{\to}e))$ to $e$, concluding the process of the $l=2$ step.

## 1.2 Nonbacktracking lower bounds (NB-LB)

Figure 2: The step-wise illustration of NB-LB on the example network.

| | $k=1$ | $k=2$ | $k=3$ | $k=4$ |
|---|---|---|---|---|
| $v_k$ | $a$ | $b$ | $c$ | $d$ |
| $\mathrm{M}(v_k)$ | $\{(1,1)\}$ | $\{(1,p)\}$ | $\{(1,p),(p,p)\}$ | $\{(p+p^2-p^3,p)\}$ |
| $\mathrm{LB}(v_k)$ | $1$ | $p$ | $p+p^2-p^3$ | $p^2+p^3-p^4$ |
| $N^+(v_k)\setminus S_0$ | $\{b,c\}$ | $\{c\}$ | $\{d\}$ | $\varnothing$ |
| $(\mathrm{LB}(v_k),\mathcal{P}'_{v_k v_l})$ to $v_l$ | $(1,p)$ to $b$ and $c$ | $(p,p)$ to $c$ | $(p+p^2-p^3,p)$ to $d$ | |
| $\sigma^-$ | $1$ | $1+p$ | $1+2p+p^2-p^3$ | $1+2p+2p^2-p^4$ |

Table 2: The values of the key variables in NB-LB on the example network in Figure 2.

In Figure 2, we show an example for the lower bound computation by NB-LB on a small network $IC(G,\mathcal{P},S_0)$ defined on an undirected graph $G=(V,E)$, where $V=\{a,b,c,d\}$, $S_0=\{a\}$, and every edge has the same transmission probability $p$. For each $k$, Table 2 shows the values of the key variables, $\mathrm{M}(v_k)$, $\mathrm{LB}(v_k)$, and $(\mathrm{LB}(v_k),\mathcal{P}'_{v_k v_l})$ for the out-neighbors $v_l \in N^+(v_k)\setminus S_0$, and shows the changes in $\sigma^-$.

We obtain MDAS from the network as follows. Since $d(S_0,a)=0$, $d(S_0,b)=d(S_0,c)=1$ and $d(S_0,d)=2$, we order the vertices as $\{v_1=a,v_2=b,v_3=c,v_4=d\}$ to satisfy that $d(S_0,v_i)\leq d(S_0,v_j)$, for every $i<j$.

NB-LB algorithm processes the nodes $\{v_1=a,v_2=b,v_3=c,v_4=d\}$ sequentially. For example, at $k=3$, node $c$ is processed. Recall that at $k=1$, node $a$ sent the message $(\mathrm{LB}(a),\mathcal{P}'_{ac})$ to node $c$, and at $k=2$, node $b$ sent the message $(\mathrm{LB}(b),\mathcal{P}'_{bc})$ to node $c$. Thus,

$$\mathrm{M}(c)=\{(\mathrm{LB}(a),\mathcal{P}'_{ac}),(\mathrm{LB}(b),\mathcal{P}'_{bc})\}=\{(1,p),(p,p)\}. \tag{11}$$

Then, it computes $\mathrm{LB}(c)$ with the function $\texttt{ProcessIncomingMsg}_{\mathrm{LB}}$.

$$\begin{aligned}\mathrm{LB}(c) &= \texttt{ProcessIncomingMsg}_{\mathrm{LB}}(\mathrm{M}(c)) \tag{12}\\ &= \mathcal{P}'_{ac}\mathrm{LB}(a)+\mathcal{P}'_{bc}\mathrm{LB}(b)(1-\mathcal{P}'_{ac})=p+p^2-p^3. \tag{13}\end{aligned}$$

Recall that $\sigma^-=1+p$, at the end of iteration $k=2$. Thus,

$$\sigma^- = 1+p+\mathrm{LB}(c)=1+2p+p^2-p^3. \tag{14}$$

Next, since $N^+(c)\setminus S_0=\{d\}$, node $c$ sends the message $(\mathrm{LB}(c),\mathcal{P}_{cd})=(p+p^2-p^3,p)$ to node $d$, concluding the process of the $k=3$ step.

## 2 Proofs of the theorems

We start by defining the following events for the independent cascade model $IC(G,\mathcal{P},S_0)$, where $G=(V,E)$ and $|V|=n$.

**Definition 10.** *For any $u,v\in V$, $l\in\{0,\ldots,n-1\}$, and $S\subseteq V$, we define*

$$\begin{aligned}A(v) &= \{v \text{ is influenced}\} \tag{15}\\ A_l(v) &= \cup_{P\in P_l(S_0\rightsquigarrow v)}\{P \text{ is open}\} \tag{16}\\ A_l(u\to v) &= \cup_{P\in P_l(S_0\rightsquigarrow u),P\not\ni v}\{P \text{ is open and edge }(u,v)\text{ is open}\} \tag{17}\\ A_{l,S}(v) &= \cup_{P\in\{P'\in P_l(S_0\rightsquigarrow v):P'\not\ni w,\forall w\in S\}}\{P \text{ is open}\}. \tag{18}\end{aligned}$$

In other words, $A_l(v)$ is the event that node $v$ is influenced by open paths of length $l$, $A_l(u\to v)$ is the event that $v$ is influenced by node $u$ with open paths of length $l+1$, i.e. there exists an open path of length $l+1$ from a seed to $v$ that ends with edge $(u,v)$, and $A_{l,S}(v)$ is the event that node $v$ is influenced by length $l$ open paths which do not include any node in $S$.

**Lemma 1.** *For any $v \in V$,*

$$p(v) \leq 1 - \prod_{l=0}^{n-1}(1 - p_l(v)). \tag{19}$$

*For any $v \in V$ and $l \in \{0, \ldots, n-1\}$,*

$$p_l(v) \leq 1 - \prod_{u \in N^-(v)}(1 - p_l(u \to v)). \tag{20}$$

*Proof.* Recall that $p(v) = \mathbb{P}(A(v))$, $p_l(v) = \mathbb{P}(A_l(v))$, and $p_l(u \to v) = \mathbb{P}(A_l(u \to v))$.

$$
\begin{align}
p(v) &= \mathbb{P}(\cup_{l=0}^{n-1} A_l(v)) \tag{21}\\
&= 1 - \mathbb{P}(\cap_{l=0}^{n-1} A_l(v)^C) \tag{22}\\
&\leq 1 - \prod_{l=0}^{n-1} \mathbb{P}(A_l(v)^C) \tag{23}\\
&= 1 - \prod_{l=0}^{n-1}(1 - p_l(v)). \tag{24}
\end{align}
$$

Equation (23) follows from the positive correlation among the events $A_l(v)^C$ for all $v \in V$. Similarly,

$$
\begin{align}
p_l(v) &= \mathbb{P}(\cup_{u \in N^-(v)} A_l(u \to v)) \tag{25}\\
&= 1 - \mathbb{P}(\cap_{u \in N^-(v)} A_l(u \to v)^C) \tag{26}\\
&\leq 1 - \prod_{u \in N^-(v)} \mathbb{P}(A_l(u \to v)^C) \tag{27}\\
&= 1 - \prod_{u \in N^-(v)}(1 - p_l(u \to v)). \tag{28}
\end{align}
$$

$\square$

**Theorem 2.** *For any independent cascade model $IC(G, \mathcal{P}, S_0)$,*

$$\sigma(S_0) \leq \sum_{v \in V}(1 - \prod_{l=0}^{n-1}(1 - \mathrm{UB}_l(v))) =: \sigma^+(S_0), \tag{29}$$

*where $\mathrm{UB}_l(v)$ is obtained recursively as in Definition 7.*

*Proof.* We provide a proof by induction. The initial condition, for $l = 0$, can be easily checked. For every $s \in S_0$, $s^+ \in N^+(s)$, $u \in V \setminus S_0$, and $v \in N^+(u)$,

$$
\begin{align}
p_0(s) = 1 &\leq \mathrm{UB}_0(s) = 1 \tag{30}\\
p_0(s \to s^+) = \mathcal{P}_{ss^+} &\leq \mathrm{UB}_0(s \to s^+) = \mathcal{P}_{ss^+} \tag{31}\\
p_0(u) = 0 &\leq \mathrm{UB}_0(u) = 0 \tag{32}\\
p_0(u \to v) = 0 &\leq \mathrm{UB}_0(u \to v) = 0. \tag{33}
\end{align}
$$

For each $l \leq L$, assume that $p_l(v) \leq \mathrm{UB}_l(v)$ and $p_l(u \to v) \leq \mathrm{UB}_l(u \to v)$ for all $u, v \in V$.

Since $p_l(s) = p_l(s \to s^+) = p_l(s^- \to s) = 0$ for every $l \geq 1$, $s \in S_0$, $s^+ \in N^+(s)$, and $s^- \in N^-(s)$, it is sufficient to show $p_{L+1}(v) \leq \mathrm{UB}_{L+1}(v)$ and $p_{L+1}(u \to v) \leq \mathrm{UB}_{L+1}(u \to v)$ for all $u \in V \setminus S_0$, and $v \in N^+(u)$,

For simplicity, for any pair of events $(A, B)$, use the notation $AB$ for $A \cap B$.

For any $v \in V \setminus S_0$,

$$p_{L+1}(v) = \mathbb{P}(\cup_{u \in N^-(v)} E_{uv} A_{L,\{v\}}(u)), \tag{34}$$

where $E_{uv}$ denotes the event that edge $(u, v)$ is open, i.e. $\mathbb{P}(E_{uv}) = \mathcal{P}_{uv}$. Thus,

$$p_{L+1}(v) \quad = \quad 1 - \mathbb{P}(\cap_{u \in N^-(v)}(E_{uv}A_{L,\{v\}}(u))^C) \tag{35}$$

$$\leq \quad 1 - \prod_{u \in N^-(v)} (1 - \mathbb{P}(E_{uv}A_{L,\{v\}}(u))) \tag{36}$$

$$= \quad 1 - \prod_{u \in N^-(v)} (1 - p_L(u \to v)) \tag{37}$$

$$\leq \quad 1 - \prod_{u \in N^-(v)} (1 - \mathrm{UB}_L(u \to v)) = \mathrm{UB}_{L+1}(v), \tag{38}$$

where Equation (36) is obtained by the positive correlation among the events $E_{uv}A_{L,\{v\}}(u)$, and Equation (38) comes from the assumption.

For any $v \in V \setminus S_0$ and $w \in N^+(v)$,

$$p_{L+1}(v \to w) \quad = \quad \mathbb{P}(E_{vw}A_{L+1,\{w\}}(v)). \tag{39}$$

$$= \quad \mathcal{P}_{vw}\mathbb{P}(A_{L+1,\{w\}}(v)) \tag{40}$$

Equation (40) follows from the independence between the events $E_{vw}$ and $A_{L+1,\{w\}}(v)$.
If $w \in N^-(v)$,

$$p_{L+1}(v \to w) \quad = \quad \mathcal{P}_{vw}\mathbb{P}(\cup_{u \in N^-(v)\setminus\{w\}}E_{uv}A_{L,\{v,w\}}(u)) \tag{41}$$

$$\leq \quad \mathcal{P}_{vw}\left(1 - \prod_{u \in N^-(v)\setminus\{w\}}(1 - \mathbb{P}(E_{uv}A_{L,\{v,w\}}(u)))\right) \tag{42}$$

$$\leq \quad \mathcal{P}_{vw}\left(1 - \prod_{u \in N^-(v)\setminus\{w\}}(1 - p_L(u \to v))\right) \tag{43}$$

$$\leq \quad \mathcal{P}_{vw}\left(1 - \prod_{u \in N^-(v)\setminus\{w\}}(1 - \mathrm{UB}_L(u \to v))\right), \tag{44}$$

Equation (43) holds, since the two events satisfy $E_{uv}A_{L,\{v,w\}}(u) \subseteq E_{uv}A_{L,\{v\}}(u)$.
Recall that, if $w \in N^-(v)$,

$$\mathrm{UB}_{L+1}(v \to w) \quad = \quad \mathcal{P}_{vw}(1 - \frac{1 - \mathrm{UB}_{L+1}(v)}{1 - \mathrm{UB}_L(w \to v)}) \tag{45}$$

$$= \quad \mathcal{P}_{vw}(1 - \prod_{u \in N^-(v)\setminus\{w\}}(1 - \mathrm{UB}_L(u \to v))). \tag{46}$$

Thus, $p_{L+1}(v \to w) \leq \mathrm{UB}_{L+1}(v \to w)$, for all $w \in N^+(v) \cap N^-(v)$.
If $w \notin N^-(v)$,

$$p_{L+1}(v \to w) \quad = \quad \mathcal{P}_{vw}\mathbb{P}(\cup_{u \in N^-(v)}E_{uv}A_{L,\{v,w\}}(u)) \tag{47}$$

$$\leq \quad \mathcal{P}_{vw}\left(1 - \prod_{u \in N^-(v)}(1 - \mathrm{UB}_L(u \to v))\right) \tag{48}$$

$$= \quad \mathcal{P}_{vw}\mathrm{UB}_{L+1}(v) = \mathrm{UB}_{L+1}(v \to w), \tag{49}$$

Hence, $p_{L+1}(v \to w) \leq \mathrm{UB}_{L+1}(v \to w)$, for all $w \in N^+(v)$, concluding the induction proof.
Finally, by Lemma 1,

$$\sigma(S_0) \quad \leq \quad \sum_{v \in V}(1 - \prod_{l=0}^{n-1}(1 - p_l(v))) \tag{50}$$

$$\leq \quad \sum_{v \in V}(1 - \prod_{l=0}^{n-1}(1 - \mathrm{UB}_l(v))) = \sigma^+(S_0). \tag{51}$$

$\square$

**Theorem 3.** *For any independent cascade model $IC(G, \mathcal{P}, S_0)$ and its directed acyclic subnetwork $IC(G', \mathcal{P}', S_0)$,*

$$\sigma(S_0) \geq \sum_{v_k \in V'} \mathrm{LB}(v_k) =: \sigma^-(S_0), \tag{52}$$

*where $\mathrm{LB}(v_k)$ is obtained recursively as in Definition 9.*

*Proof.* We provide a proof by induction. For any $v_k \in V'$, let $A(v_k)$ be the event that node $v_k$ is influenced in MDAS $IC(G', \mathcal{P}', S_0)$, and for every edge $(v_j, v_k)$, let $E_{v_j, v_k}$ be the event that edge $(v_j, v_k)$ is open, i.e. $\mathbb{P}(E_{v_j, v_k}) = \mathcal{P}'_{v_j v_k}$. Recall that $p(v_k) = \mathbb{P}(A(v_k))$.

The initial condition $k = 1$ holds, since $p(v_1) = 1 \geq \mathrm{LB}(v_1) = 1$ ($v_1$ is a seed).

For every $k \leq K$, assume $p(v_k) \geq \mathrm{LB}(v_k)$.

For the node $v_{K+1}$,

$$p(v_{K+1}) = \mathbb{P}(\cup_{v_j \in N^-(v_{K+1})} E_{v_j v_{K+1}} A(v_j)). \tag{53}$$

We re-label vertices in $N^-(v_{K+1}) = \{u_1, \ldots, u_m\}$ where $m = \text{in-deg}(v_{K+1})$, and let $\mathcal{Q}_{iK+1} = \mathcal{P}'_{u_i v_{K+1}}$. Then, for any integer $m' \leq m$,

$$
\begin{align}
p(v_{K+1}) &= \mathbb{P}(\cup_{i=1}^{m} E_{u_i v_{K+1}} A(u_i)) \tag{54} \\
&\geq \mathbb{P}(\cup_{i=1}^{m'} E_{u_i v_{K+1}} A(u_i)) \tag{55} \\
&\geq \sum_{i=1}^{m'} \mathbb{P}(E_{u_i v_{K+1}} A(u_i)) - \sum_{i=1}^{m'} \sum_{j=1}^{i-1} \mathbb{P}(E_{u_i v_{K+1}} A(u_i) E_{u_j v_{K+1}} A(u_j)) \tag{56} \\
&= \sum_{i=1}^{m'} \mathcal{Q}_{iK+1} \mathbb{P}(A(u_i)) - \sum_{i=1}^{m'} \sum_{j=1}^{i-1} \mathcal{Q}_{iK+1} \mathcal{Q}_{jK+1} \mathbb{P}(A(u_i) A(u_j)) \tag{57} \\
&\geq \sum_{i=1}^{m'} \mathcal{Q}_{iK+1} \mathbb{P}(A(u_i))(1 - \sum_{j=1}^{i-1} \mathcal{Q}_{jK+1}). \tag{58}
\end{align}
$$

Equation (56) follows from the principle of inclusion and exclusion. Equation (57) results from the Independence between the event that an edge ending with $v_{K+1}$ is open and the event that a node $v_i$ is influenced where $i < K + 1$. Equation (58) holds since $\mathbb{P}(A(u_i)) \geq \mathbb{P}(A(u_i) A(u_j))$.

Now, define $m^* = \max\{m' \leq m : \sum_{j=1}^{m'-1} \mathcal{Q}_{jK+1} \leq 1\}$. Then,

$$
\begin{align}
p(v_{K+1}) &\geq \sum_{i=1}^{m^*} \mathcal{Q}_{iK+1} \mathbb{P}(A(u_i))(1 - \sum_{j=1}^{i-1} \mathcal{Q}_{jK+1}) \tag{59} \\
&\geq \sum_{i=1}^{m^*} \mathcal{Q}_{iK+1} \mathrm{LB}(u_i)(1 - \sum_{j=1}^{i-1} \mathcal{Q}_{jK+1}) \tag{60} \\
&= \mathrm{LB}(v_{K+1}). \tag{61}
\end{align}
$$

Equation (60) follows since $1 - \sum_{j=1}^{i-1} \mathcal{Q}_{jK+1} \geq 0$ for all $i \leq m^*$ by the definition of $m^*$. Thus, $p(v_i) \geq \mathrm{LB}(v_i)$ for all $v_i \in V'$, concluding the induction proof.
Finally,

$$
\begin{align}
\sigma(S_0) &\geq \sum_{i=1}^{n} p(v_i) \tag{62} \\
&\geq \sum_{i=1}^{n} \mathrm{LB}(v_i) = \sigma^-(S_0). \tag{63}
\end{align}
$$

Equation (62) holds since its right hand side equals to the influence of the MDAS, $IC(G', \mathcal{P}', S_0)$. $\square$

# 3 Tunable nonbacktracking bounds

We present here the parametrized algorithms for NB-UB and NB-LB described in Section 3.3.

**Tunable nonbacktracking upper bounds (tNB-UB)**: The algorithm inputs the parameter $t$, which indicates the maximum length of the paths that the algorithm considers in order to compute the exact, rather than the upper bound on, probability of influence. That is, the algorithm computes $p_{\leq t}(u)$ that node $u$ is influenced by an open path whose length is less than or equal to $t$.

$$p_{\leq t}(u) \quad = \quad \mathbb{P}(\cup_{P \in \{\cup_{i=0}^{t} P_i(S_0 \to u)\}} \{P \text{ is open}\}). \tag{64}$$

Then, we start (non-parameterized) NB-UB algorithm from $l = t + 1$ with the new initial conditions: for all $u \in V$ and $v \in N^+(u)$,

$$\mathrm{UB}_t(u) = p_{\leq t}(u) \tag{65}$$

$$\mathrm{UB}_t(u \to v) = p_t(u \to v) \tag{66}$$

Finally, the upper bound by tNB-UB is computed as $\sum_{v \in V}(1 - \prod_{l=t}^{n-1}(1 - \mathrm{UB}_l(v)))$.

---

**Algorithm** Tunable NB-UB (tNB-UB)

---

    **parameter:** non-negative integer $t \leq n - 1$
    **Initialize:** $\mathrm{UB}_t(v) = 0$ for all $t \leq l \leq n - 1$ and $v \in V$
    **for** $u \in V$ **do**
        $\mathrm{UB}_t(u) = p_{\leq t}(u)$
        **for** $v \in N^+(u) \setminus S_0$ **do**
            **if** $p_t(u \to v) > 0$ **then**
                $S_{t+1}.\mathrm{insert}(v)$
                $\mathrm{M}_{\mathrm{next}}(v).\mathrm{insert}(u, p_t(u \to v))$
    **for** $l = t + 1$ **to** $n - 1$ **do**
        **for** $u \in S_l$ **do**
            $\mathrm{M}_{\mathrm{curr}}(u) = \mathrm{M}_{\mathrm{next}}(u)$
            Clear $\mathrm{M}_{\mathrm{next}}(u)$
            $\mathrm{UB}_l(u) = \mathtt{ProcessIncomingMsg_{UB}}(\mathrm{M}_{\mathrm{curr}}(u))$
        **for** $u \in S_l$ **do**
            **for** $v \in N^+(u) \setminus S_0$ **do**
                $S_{l+1}.\mathrm{insert}(v)$
                **if** $v \in \mathrm{M}_{\mathrm{curr}}(u)$ **then**
                    $\mathrm{UB}_l(u \to v) = \mathtt{GenerateOutgoingMsg_{UB}}(\mathrm{M}_{\mathrm{curr}}(u)[v], \mathrm{UB}_l(u), \mathcal{P}_{uv})$
                    $\mathrm{M}_{\mathrm{next}}(v).\mathrm{insert}((u, \mathrm{UB}_l(u \to v))).$
                **else**
                    $\mathrm{UB}_l(u \to v) = \mathtt{GenerateOutgoingMsg_{UB}}(0, \mathrm{UB}_l(u), \mathcal{P}_{uv})$
                    $\mathrm{M}_{\mathrm{next}}(v).\mathrm{insert}((u, \mathrm{UB}_l(u \to v))).$
    **Output:** $\mathrm{UB}_l(u)$ for all $l = \{t, t + 1, \ldots, n - 1\}, u \in V$

---

**Tunable nonbacktracking lower bounds (tNB-LB)**: We first order the vertex set as $V' = \{v_1, \ldots, v_n\}$, which satisfies $d(S_0, v_i) \leq d(S_0, v_j)$, for every $i < j$. Given a non-negative integer parameter $t \leq n$, we obtain a $t$-size subnetwork $IC(G[V_t], \mathcal{P}[V_t], S_0 \cap V_t)$, where $G[V_t]$ is the vertex-induced subgraph which is induced by the set of nodes $V_t = \{v_1, \ldots, v_t\}$, and $\mathcal{P}[V_t]$ is the corresponding transmission probability matrix. For each $v_i \in V_t$, we compute the exact probability $p_t(v_i)$ that node $v_i$ is influenced in the subnetwork $IC(G[V_t], \mathcal{P}[V_t], S_0 \cap V_t)$. Then, we start (non-parameterized) NB-LB algorithm from $k = t + 1$ with the new initial condition: for all $k \leq t$,

$$\mathrm{LB}(v_k) = p_t(v_k). \tag{67}$$

Finally, tNB-LB computes the lower bound as $\sum_{v_k \in V'} \mathrm{LB}(v_k)$.

---
**Algorithm** Tunable NB-LB (tNB-LB)
---
**parameter:** non-negative integer $t \leq n$
**Initialize:** $\sigma^- = 0$
**for** $k = 1$ **to** $t$ **do**
    $\text{LB}(v_k) = p_t(v_k)$
    $\sigma^- \mathrel{+}= \text{LB}(v_k)$
    **for** $v_i \in \{N^+(v_k) \cap \{v_j : j > t\}\}$ **do**
        $\text{M}(v_i).\text{insert}((\text{LB}(v_k), \mathcal{P}'_{v_k v_i}))$
**for** $k = t + 1$ **to** $n$ **do**
    $\text{LB}(v_k) = \texttt{ProcessIncomingMsg}_{\text{LB}}(\text{M}(v_k))$
    $\sigma^- \mathrel{+}= \text{LB}(v_k)$
    **for** $v_i \in N^+(v_k) \setminus S_0$ **do**
        $\text{M}(v_i).\text{insert}((\text{LB}(v_k), \mathcal{P}'_{v_k v_i}))$
**Output:** $\sigma^-$
---

**Experimental results**: In Figure 3a, we show tNB-UB on a sample network. We consider a 3-regular network with 100 nodes and a single seed. Since the NB-UB gives a tight bound on $p < 0.4$, we plot tNB-UB on $p \in (0.4, 0.5)$ where it shows some improvements with small $t$.

In Figure 3b, we present tNB-LB on a scale-free network with 3000 nodes, $\alpha = 2.5$, and a single seed. We compare tNB-LB with various choices of $t \in \{1, 12, 100, 300\}$, and tNB-LB approaches the MC estimation as $t$ grows.

Figure 3: (a) NB-UB, tNB-UB with $t = 3$, and MC estimation with 10000 simulations on a 3-regular network with 100 nodes.
(b) NB-LB, tNB-LB with various $t \in \{12, 100, 300\}$, and MC estimation with 3000000 simulations on a scale-free network with 3000 nodes.