[Reviews · NeurIPS 2017]

Reviewer 1



Nonbacktracking Bounds on the Influence in Independent Cascade Models develops bounds on the expected number of nodes that eventually would be "infected" from a seminal set. The bounds computations depend on a recursive computation that is of lower order than a Monte Carlo computation to the same end, and obtains an estimate of variance at no additional computational cost. The propagation ends when all nodes have been "touched", with the probability diminishing of becoming infected as the length of the path from the source nodes increases. The recursion takes care not to propagate back to the previous node from which the infection came, hence the non-backtracking property. A more exact method would exclude not only immediate backtracking, but any loop formed by the propagation path, as the authors recognize. Not being specifically versed in this literature, the paper appears correct and novel. It is apparently written for a specific audience who is familiar with current approaches to this problem.

Reviewer 2



The paper develops upper and lower bounds, using correlation inequalities, on the value of the influence function (expected influenced set size) under the Independent Cascade model. The work is a clear improvement on the upper bounds of Lemonnier et al., NIPS 2014. The theoretical results are a solid contribution towards a bound-based analysis of influence computation and maximization, though the evaluation could be improved to better evaluate and communicate the impact of the results. 1) As future work building on this line of bounds, It would be interesting to see work connecting them to actual influence maximization, e.g. running a greedy IM procedure, and comparing the seed sets under greedy IM vs. seed sets returned by greedy maximization over the UB or LB measures. It would have been nice to see some dialogue with maximization as part of the work, but it would have been difficult to fit into the short format. 2) In the Experimental Results (Section 4), it would be preferable to see the evaluation decomposed over the two main sources of randomness being studied, the graph and the seed node {s]. It would be interesting to dig into properties of the seed nodes for which the bounds are tighter or looser (degree? Centrality? etc.) It would also be preferable to see an evaluation of larger seed sets, as part of the difficulty with computing influence is interactions between seed nodes. 3) An obvious omission is the study of real world graphs, which have higher degrees and contain many more triangles than any of the simulated graphs considered in the paper. The higher degrees make backtracking steps in random walks less frequent. The triangles and other cycles generally impede message passing algorithms. It’d be worth evaluating and commenting on these considerations. 4) The state of the art for fast IM computation with approximation guarantees is, I believe, (Tang-Xiao-Shi, SIGMOD 14). Worth including in the citations to IM papers. Furthermore, the use of the word “heuristic” on Line 20 is usually reserved for algorithms without formal guarantees. The methods in the citation block [10, 17, 3, 7, 20] almost all feature formal guarantees for IM in terms of worst-case approximation ratios (some in probabilistic senses). To call all these papers “heuristic” feels overly dismissive. As a related comment on the literature review, on Line 32 "A few exceptions include” makes it sound like there is going to be more than one exception, but only one is given. Fine to rewrite to only expect on exception, better to give more cites.

Reviewer 3



Finding influence in the IC model is a #P hard problem, and as such one cannot expect to solve it exactly and this motivates the search for approximating influence efficiently. Monte-Carlo methods may be used, but they may be computationally expensive based on the graph characteristics. This paper looks to obtain lower and upper bounds on the influence, which (if tight) serve as a proxy for the actual influence, and otherwise serve as a way to bound the variance of the influence which may then be useful for a Monte-Carlo algorithm. The paper presents both upper and lower bounds, with the main idea being that influence is given by the counting the probabilities of all simple paths and summing them up, so an upper bound would "over-count" and a lower bound would "under-count" in some suitable way. For the upper bound, the paper proposes using non-backtracking paths and gives an efficient algorithm that counts the path weights for all such paths. For the lower bound, a directed acyclic subgraph is constructed and influence for this subgraph is used to bound the influence of the actual graph. The paper is technically correct, and reasonably well-written. The upper bound and the algorithm to efficiently compute it is novel. One negative aspect is that is no theoretical study of tightness of these bounds. It will be good to understand for what kinds of graphs is the upper bound tight. (Presumably the lower bound is tight for DAGs.)